# SEMI-SUPERVISED LEARNING WITH MULTI-DOMAIN SENTIMENT WORD EMBEDDINGS

## ABSTRACT

Word embeddings are known to boost performance of many NLP tasks such as text classification, meanwhile they can be enhanced by labels at the document level to capture nuanced meaning such as sentiment and topic. Can one combine these two research directions to benefit from both? In this paper, we propose to jointly train a text classifier with a label-enhanced and domain-aware word embedding model, using an unlabeled corpus and only a few labeled data from non-target domains. The embeddings are trained on the unlabed corpus and enhanced by pseudo labels coming from the classifier, and at the same time are used by the classifier as input and training signals. We formalize this symbiotic cycle in a variational Bayes framework, and show that our method improves both the embeddings and the text classifier, outperforming state-of-the-art domain adaptation and semi-supervised learning techniques. We conduct detailed ablative tests to reveal gains from important components of our approach. The source code and experiment data will be publicly released.

## 1 INTRODUCTION

Widely used word embeddings (Mikolov et al., 2013b; Pennington et al., 2014) are generally trained from unlabeled corpora, only making use of the distribution of co-occurring context words to capture syntactic and semantic similarities. It is known that other types of information, such as document labels and sentiment polarities, can further enhance the embeddings to give focus to specific aspects of meaning that are not easily extracted otherwise (Yu & Dredze, 2014; Xu et al., 2014; Sun et al., 2015; Tang et al., 2016; Shi et al., 2018; Ye et al., 2018). For example, the word "*trash*" is semantically related to "*dumpster*", but its sentiment might be closer to "*horrible*" or "*nonsense*". Proper use of different embeddings is beneficial to downstream tasks and crucial to understanding human language.

However, to train enhanced embeddings usually requires a large amount of additional labels, which can be costly if annotated manually. To automatically annotate text documents with labels is itself a challenging NLP task, for which word embeddings can be extremely helpful (Jin et al., 2016). Therefore, it is well-motivated to combine these two inter-dependent research directions.

In this paper, we show that it is possible to jointly train a label-enhanced and domain-aware embedding model with a highly accurate text classifier, given only an unlabeled corpus and a few labeled data from non-target domains. This technique drastically reduces the cost of annotation for training label-enhanced embeddings, and at the same time greatly helps adapt text classifiers into new domains.

To be more specific, we are given a corpus of user reviews for different products and services (i.e., domains), wherein only a small portion is annotated with sentiment labels; some entire domains may consist of unlabeled reviews. We train word embeddings on this corpus, with domain information and latent sentiment labels integrated into the model; meanwhile, a classifier is trained to predict the latent sentiment, using the embeddings as input. We expect three advantages in this approach: first, a joint classifier can produce pseudo-labels for unlabeled data with high accuracy, which help train label-enhanced embeddings on a large unlabeled corpus; second, the embeddings used as input to the classifier capture sentiment semantics that is general across domains, which helps domain adaptation; third, the sentiment-aware embeddings may even provide training signals to the classifier, as a text review containing more "positive words" is likely to be positive. We formulate all these intuitions in a variational Bayes framework, so that one can freely design classifiers and embeddings.

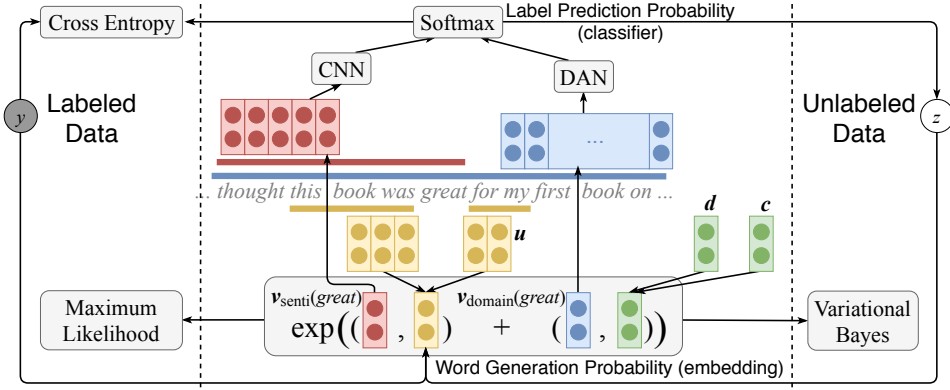

Figure 1: We jointly train a sentiment classifier with a sentiment- and domain-aware embedding model, using both labeled and unlabeled data. When sentiment label is observed, our model is trained with the usual cross entropy and maximum likelihood objectives; for unlabeled data, it uses pseudo labels produced by the sentiment classifier, and a variational Bayes objective.

Empirically, we show that our method both improves the sentiment classifier and enhances the word embeddings to be more sentiment focused (Sec.4.1, 4.2). We achieve state-of-the-art compared to previous domain adaptation and semi-supervised learning techniques (Sec.4.1), and by detailed ablative tests we show that (Sec.4.3): (i) a large unlabeled corpus combined with better classifier leads to better sentiment-aware embeddings; (ii) additional knowledge such as sentiment labels and domain information improves the classifier; and (iii) sentiment-aware embeddings have the potential to be used as training signals to the classifier and indeed improve performance in some cases.

## 2 OUR MODEL

Our dataset consists of $(D, c, y)$-tuples, where $D$ is a text document, $c \in C$ indicates the domain or "category" of the document, and $y \in L$ is the label we want to annotate $D$ with. For example, $D$ can be a user review, $c$ is the category of the product (e.g., `books` or `electronics`), and $y$ is the sentiment label of the review (e.g. `positive` or `negative`). We assume that domain $c$ is always observed, but label $y$ can be unknown. When $y$ is not observed, we denote the corresponding latent variable as $z$. In semi-supervised learning, we train a text classifier on labeled data and further exploit a generative description of unlabeled data to improve upon the supervised classifier.

### 2.1 VARIATIONAL BAYES SEMI-SUPERVISED LEARNING

We train a classifier $q_\phi(y \mid D, c)$ to model the probability of a given document $D$ being annotated with label $y$. In addition, we propose a generative model $p_\theta(D \mid y, c)$ (which is a label-enhanced word embedding model) in this work to estimate the probability of document $D$ given the label $y$ and domain $c$. Here, $\phi$ and $\theta$ are model parameters and their specific designs are discussed later. Note that $p_\theta(D \mid y, c)$ depends on $y$ (i.e. label-enhanced); so only if the label $y$ is observed, can $p_\theta(D \mid y, c)$ be directly optimized with the usual maximum likelihood objective. When the label is unknown, the standard practice of Bayesian inference will be to assume a prior $p(z \mid c)$, calculate the marginal $p_\theta(D \mid c) = \sum_z p_\theta(D \mid z, c)p(z \mid c)$ and maximize it on the unlabeled data. However, this might not actually work in practice, as in our experiments naively maximizing the marginal likelihood on a large unlabeled corpus results in models that infer latent labels as either all positive or all negative (Sec.4.3). Fortunately, the classifier $q_\phi(y \mid D, c)$ may come to help and provide a good estimate for the latent label; the idea of variational Bayes (Kingma et al., 2014) is to use $q_\phi(z \mid D, c)$ to approximate the posterior $p_\theta(z \mid D, c)$. So we start from the Bayesian inference

$$p_\theta(D \mid c) = \frac{p_\theta(D \mid z, c) \, p(z \mid c)}{p_\theta(z \mid D, c)}, \tag{1}$$

and take $\log(\cdot)$ of both sides and trivially introduce the term $\log q_\phi(z \mid D, c)$:

$$\log p_\theta(D \mid c) = \log \frac{q_\phi(z \mid D, c)}{p_\theta(z \mid D, c)} + \log p_\theta(D \mid z, c) + \log p(z \mid c) - \log q_\phi(z \mid D, c). \qquad (2)$$

Then, we take the expectation $\mathbb{E}_{q_\phi(z \mid D,c)}[\cdot]$ of both sides, and recall that KL-divergence is non-negative:

$$\mathbb{E}_{q_\phi(z \mid D,c)}\Big[\log \frac{q_\phi(z \mid D, c)}{p_\theta(z \mid D, c)}\Big] = KL\big[q_\phi(z \mid D, c) \,||\, p_\theta(z \mid D, c)\big] \geq 0.$$

Thus, we have

$$\log p_\theta(D \mid c) \geq \sum_{z \in L} q_\phi(z \mid D, c)\big(\log p_\theta(D \mid z, c) + \log p(z \mid c) - \log q_\phi(z \mid D, c)\big), \qquad (3)$$

and we obtained a lower bound for the log-likelihood $\log p_\theta(D \mid c)$ that involves $q_\phi(z \mid D, c)$. The variational Bayes objective modifies the usual maximum likelihood estimator by replacing $\log p_\theta(D \mid c)$ with this lower bound. The weighted sum in Equation (3) with weight $q_\phi(z \mid D, c)$ suggests that "pseudo sentiment labels" are drawn from the distribution $q_\phi(z \mid D, c)$, and the log-likelihood of the embedding model $\log p_\theta(D \mid z, c)$ is maximized according to these pseudo labels. We note that our application of the variational Bayes is slightly different from typical situations, in which those posteriors are intractable and approximation is necessary; in contrast, our problem allows precise Bayesian inference, nevertheless we involve a classifier $q_\phi(z \mid D, c)$ in order to benefit from more freedom of design, because some discriminative models do not enjoy a generative description, yet they are strong classifiers and outperform Bayesian inference of good generative models. In our experiments, we show that a better classifier indeed leads to better training of our Bayesian model, and variational Bayes can outperform pure Bayesian inference (Sec.4.3).

On the other hand, the expression $\log p_\theta(D \mid z, c) + \log p(z \mid c) - \log q_\phi(z \mid D, c)$ in Equation (3) is the difference between Bayes inference and the classifier prediction; it might serve as training signals to the classifier if the generative model is good enough to provide strong Bayes predictions. In this sense, we are toward the ultimate goal of semi-supervised learning to train a classifier from unlabeled data. In Sec.4.3, we empirically investigate the effect of this training signal.

In practice, the training signal coming from unlabeled data is noisy, so we have to over-sample the labeled data to enforce appropriate training of the classifier. Concretely, each time the classifier is trained on an unlabeled document (using the variational Bayes objective), we additionally train it on a labeled random sample as well (using the usual cross entropy loss). Furthermore, the learning rates for gradient updates from unlabeled data are set smaller (see Appendix for details).

## 2.2 MULTI-DOMAIN SENTIMENT WORD EMBEDDING

In this work, a document $D = (w_1, \ldots, w_n)$ is regarded as a sequence of words, and its likelihood is calculated from generative probabilities of words:

$$\log p_\theta(D \mid y, c) = \frac{1}{n} \sum_{i=1}^{n} \log p_\theta(w_i \mid y, c). \qquad (4)$$

Our word generation model is based on the CBOW embedding (Mikolov et al., 2013a). Recall that in CBOW, every word $w$ is assigned a context vector $\boldsymbol{u}(w)$ and a target vector $\boldsymbol{v}(w)$, and the generative probability of each word is given by:

$$p(w_i) \propto \exp\big(\boldsymbol{v}(w_i) \cdot \sum_{0 < |j-i| \leq \delta} \boldsymbol{u}(w_j)\big).$$

Here, $\delta$ is the size of a context window. Next, we integrate sentiment $y$ and domain $c$ into this model.

### 2.2.1 SENTIMENT LABEL

In review text, word usage depends not only on the surrounding context, but also on the overall sentiment polarity. For example, the most likely word following the context "*this product is*" should be drastically different between positive and negative reviews. We model this intuition by applying an

affine transformation to context vectors according to the sentiment. Concretely, each sentiment $y \in L$ is assigned a matrix $\boldsymbol{M}(y)$ and a vector $\boldsymbol{b}(y)$, and we model sentiment-aware word generation as:

$$p(w_i \,|\, y) \propto \exp\left(\boldsymbol{v}_{\text{senti}}(w_i) \cdot \left(\boldsymbol{M}(y)\big(\sum_{0<|j-i|\leq\delta} \boldsymbol{u}(w_j)\big) + \boldsymbol{b}(y)\right)\right). \tag{5}$$

Here, $\boldsymbol{v}_{\text{senti}}$ is the sentiment-aware word embedding. In experiments, we will show that $\boldsymbol{v}_{\text{senti}}$ focuses more on the sentiment aspect of word meaning (Sec.4.2); for instance, $\boldsymbol{v}_{\text{senti}}(\textit{trash})$ should be more similar to $\boldsymbol{v}_{\text{senti}}(\textit{horrible})$ and $\boldsymbol{v}_{\text{senti}}(\textit{nonsense})$, than to $\boldsymbol{v}_{\text{senti}}(\textit{dumpster})$.

### 2.2.2 DOMAIN INFORMATION

The sentiment-aware word embedding $\boldsymbol{v}_{\text{senti}}$ is intended to generally capture sentiment across different documents and domains; in contrast, we use $\boldsymbol{v}_{\text{domain}}$ to model semantics through domain- and document-specific distributions. Concretely, we assign a unique vector $\boldsymbol{d}(D)$ to each unique document $D$, and a vector $\boldsymbol{c}(c)$ to domain $c$. The domain-specific word generation probability is given by:

$$p(w \,|\, c) \propto \exp\left(\boldsymbol{v}_{\text{domain}}(w) \cdot \big(\boldsymbol{c}(c) + \boldsymbol{d}(D)\big)\right). \tag{6}$$

Our experiments suggest that, $\boldsymbol{v}_{\text{domain}}$ gives more focus on domains or topics of words; for example, $\boldsymbol{v}_{\text{domain}}(\textit{books})$ is more similar to $\boldsymbol{v}_{\text{domain}}(\textit{author})$ and $\boldsymbol{v}_{\text{domain}}(\textit{read})$, which are related to the "books" topic, than to $\boldsymbol{v}_{\text{domain}}(\textit{novels})$, which specifies a "fiction" topic (Sec.4.2).

All set, our multi-domain sentiment word embedding is modeled as

$$p_\theta(w_i \,|\, y, c) = p(w_i \,|\, y)\, p(w_i \,|\, c), \tag{7}$$

with model parameters $\theta = \{\boldsymbol{u}, \boldsymbol{M}, \boldsymbol{b}, \boldsymbol{v}_{\text{senti}}, \boldsymbol{v}_{\text{domain}}, \boldsymbol{c}, \boldsymbol{d}\}$. In this work, we fix the dimension of all embeddings to 256, and the context window size $\delta$ is drawn every time from a Poisson distribution of mean 2.5. Further, for each target word, we distinguish context words on its left side from the right side. Following Mikolov et al. (2013b), we adopt the negative sampling optimization (Mnih & Kavukcuoglu, 2013; Gutmann & Hyvärinen, 2012) for training embeddings, maximizing the following objective for each word $w_i$ with $k = 3$ noise words (denoted $\varpi$), drawn from a noise distribution, Noise="the unigram distribution to the power of 0.75":

$$\ln\frac{p_\theta(w_i \,|\, y, c)}{k + p_\theta(w_i \,|\, y, c)} + \sum_{\varpi \sim \text{Noise}} \ln\frac{k}{k + p_\theta(\varpi \,|\, y, c)}. \tag{8}$$

### 2.3 SENTIMENT CLASSIFIER

Our design for the classifier $q_\phi(y \,|\, D, c)$ consists of a generic part and a domain-specific part. The generic part uses a Convolutional Neural Network (CNN) (Lecun et al., 1998; Collobert et al., 2011; Kim, 2014) to predict sentiment from distinctive short phrases (e.g. "*thought this book was great*"). It takes the sentiment-aware embedding $\boldsymbol{v}_{\text{senti}}$ as input, in order to generalize across different domains and different phrases of similar sentiment and semantics. On the other hand, the domain-specific part takes the domain-focused embedding $\boldsymbol{v}_{\text{domain}}$ as input and is separately trained for each domain, using a Deep Averaging Network (DAN) (Iyyer et al., 2015) to capture correlations between sentiment and topics that are usually domain-specific. For example, topics related to "*broken*" are strongly negative in electronics domain (e.g. "*earphone is broken*"), but are less so in books domain (as in a story about "*broken friendship*", or a book well-organized that "*broken into subsections*"). DAN feeds the average of embeddings of all words in a document to a multi-layer perceptron, and is known as a strong baseline for text classification despite ignoring word order (Iyyer et al., 2015). It is also demonstrated in Tian et al. (2017) that, by averaging word embeddings, the common information encoded across all words is reinforced. Thus, we expect DAN to extract overall topics of a document, rather than specific sentiment words. Complete descriptions of our classifier are given in Appendix; an illustration of our model is presented in Figure 1. Formally, we define

$$q_\phi(y \,|\, D, c) \propto \exp\left(\boldsymbol{q}_{\text{gen}}(y) \cdot \boldsymbol{f}_{\text{CNN}}(D) + \boldsymbol{q}_{\text{spec}}(y; c) \cdot \boldsymbol{f}_{\text{DAN}}(D; c)\right), \tag{9}$$

where $\boldsymbol{q}_{\text{gen}}$ and $\boldsymbol{q}_{\text{spec}}$ are generic and domain-specific weight vectors, and $\boldsymbol{f}_{\text{CNN}}$ and $\boldsymbol{f}_{\text{DAN}}$ the CNN- and DAN-extracted feature vectors, respectively. Sharing the embeddings $\boldsymbol{v}_{\text{senti}}$ and $\boldsymbol{v}_{\text{domain}}$ with our classifier is another semi-supervised learning technique, besides the variational Bayes objective.

### 2.4 Domain-specific Prior

We also model the prior $p(z \mid c)$ in Equation (3). This is only used in the variational Bayes objective and trained from unlabeled data. Our preliminary experiments suggest that training this prior is better than fixing $p(z \mid c)$ to a uniform distribution. We set

$$p(z \mid c) \propto \exp\big(\pi_{\text{gen}}(z) + \pi_{\text{spec}}(z; c)\big), \qquad (10)$$

where $\pi_{\text{gen}}$ and $\pi_{\text{spec}}$ are trained parameters.

## 3 Related Work

Kingma et al. (2014) proposed to use variational Bayes approximation for semi-supervised learning with generative models. The formalization of our variational Bayes objective is in fact one of the very specific cases. However, the main concern regarding variational Bayes so far has been around the Variational Auto-Encoder (Kingma & Welling, 2013), in which the latent label space is continuous and the motivation comes from the intractability of precise Bayesian inference. It is not obvious whether the approximation is still beneficial in our case, where the latent space is finite and precise Bayesian inference is possible. Our motivation is to combine a classifier with a label-enhanced embedding model. Besides, our embeddings are used as input to the classifier, which is an additional technique beyond variational Bayes.

Various methods have been proposed to enhance word embeddings by linguistic resources (Yu & Dredze, 2014), knowledge graphs (Xu et al., 2014), or document labels (Sun et al., 2015) *etc.*; many of them are evaluated intrinsically in word similarity or analogy tasks. Sentiment-aware embeddings (Maas et al., 2011; Labutov & Lipson, 2013; Tang et al., 2016; An et al., 2018; Shi et al., 2018; Ye et al., 2018) are shown useful to sentiment analysis, but most of them are learned from existing sentiment lexicons or labels. We are not aware of any previous work that jointly trains sentiment-aware embeddings with a sentiment classifier, and makes use of an unlabeled corpus to improve both. Another line of research is to train general embeddings that can apply to several tasks and domains (Subramanian et al., 2018; Peters et al., 2018), wherein strong empirical results have been reported; still, our experiments will show that we can outperform the state-of-the-art methods in cross-domain sentiment classification tasks, leveraging a much smaller corpus.

Domain adaptation of sentiment classifier is an active research topic. Many approaches explore the idea of separating and extracting general vs. domain-specific features (Daumé III, 2007; Louizos et al., 2015; Kim et al., 2016; Ganin et al., 2016; Bousmalis et al., 2016; Liu et al., 2017; Zhao et al., 2017; Chen & Cardie, 2018); some of them will be compared to our model in the experiments. In addition, one can use linguistic insights to bootstrap a domain-specific sentiment lexicon (Bollegala et al., 2011; Wu & Huang, 2016; Mudinas et al., 2018), but traditionally these and other methods (Blitzer et al., 2007; Mansour et al., 2008; Duan et al., 2009; Pan et al., 2010; Yoshida et al., 2011; Chen et al., 2012; Saito et al., 2017; Ruder & Plank, 2018; Peng et al., 2018) are applied to unigram and bigram features, ignoring further sequential information. Recent developments adopt embeddings and sentence encoders (Li et al., 2018; Dong & de Melo, 2018; Ziser & Reichart, 2018); from which we choose a strong model and will compare it with our method.

## 4 Experiments

For our evaluation, the Multi-Domain Sentiment Dataset[1] consists of user reviews for products that fall into four categories: `books`, `dvd`, `electronics`, and `kitchen`; and the Skytrax User Reviews Dataset[2] consists of air service reviews, divided into `airline`, `airport`, `lounge`, and `seat`. The statistics is shown in Table 1. Each review document assumes a sentiment label, either `positive` or `negative`; except that a large portion of the Multi-Domain Sentiment Dataset is unlabeled[3]. We applied tokenization, sentence splitting, lower-casing to the review text, and filtered

---

[1]`http://www.cs.jhu.edu/~mdredze/datasets/sentiment/index2.html`

[2]`https://github.com/quankiquanki/skytrax-reviews-dataset`

[3]The unlabeled documents are assigned review scores that can be converted to sentiment labels. We randomly selected 2000 documents (converted to 1000 positive and negative each) from the `books` domain as development set, used for tuning hyper-parameters of our model. We use the same set of hyper-parameters for all experiments, and the converted labels are never used elsewhere.

| | product reviews | | | | service reviews | | | |
| --- | --- | --- | --- | --- | --- | --- | --- | --- |
| | books | dvd | electr. | kitchen | airline | airport | lounge | seat |
| positive | 993 | 995 | 987 | 996 | 22,080 | 3,914 | 816 | 453 |
| negative | 963 | 959 | 978 | 981 | 19,281 | 13,783 | 1,445 | 793 |
| unlabeled | 5,473 | 29,270 | 12,400 | 15,489 | - | - | - | - |

Table 1: Number of unique documents from different domains in the Multi-Domain Sentiment Dataset (products) and the Skytrax User Reviews Dataset (services).

| | books | dvd | electr. | kitchen | airline | airport | lounge | seat |
| --- | --- | --- | --- | --- | --- | --- | --- | --- |
| Ours | **82.57** | 82.72 | **84.51** | **86.86** | 83.65 | 66.17 | **73.87** | **84.70** |
| PBLM+CNN | 80.62 | 79.17 | 82.86 | 82.85 | **84.60** | **73.98** | 72.44 | 74.70 |
| PBLM+LSTM | 76.07 | 78.56 | 74.21 | 80.01 | 82.05 | **73.98** | 72.18 | 70.94 |
| ELMo+CNN | **82.99** | **84.38** | 83.27 | **86.63** | 81.60 | 68.93 | 69.38 | 83.06 |
| MAN | 74.28 | 73.50 | 79.35 | 80.63 | 76.33 | 66.58 | 68.02 | 72.14 |
| VFAE | 71.31 | 71.34 | 71.63 | 77.23 | 66.92 | 50.25 | 64.68 | 73.42 |
| GloVe+DAN | 74.54 | 75.74 | 79.60 | 80.12 | 75.07 | 68.94 | 72.00 | 71.27 |
| DAN:DANN | 73.41 | 74.82 | 78.58 | 80.17 | 72.90 | 67.68 | 64.31 | 61.24 |
| CNN:DANN | 74.28 | 76.41 | 79.19 | 80.02 | 73.07 | 65.64 | 71.03 | 78.73 |
| Random+CNN | 76.28 | 79.32 | 82.10 | 83.00 | 75.64 | 66.86 | 71.21 | 75.82 |
| Random+DAN | 73.82 | 73.18 | 79.20 | 79.41 | 76.60 | 65.10 | 70.45 | 74.09 |

Table 2: Accuracy (%) of sentiment classification in different target domains.

out punctuation. We also resolved all duplicated documents, reducing the number of documents by up to $4\%$ in some domains[4]. Our source code and experimental data will be publicly released.

In our setting, we train a sentiment classifier from multiple source domains and completely new targets, without labeled data from the target domains. Concretely, we follow Wu & Huang (2016) to select one of the four domains of product reviews (e.g. books) in turn as target domain, train our model on unlabeled product reviews and all labeled data from the remaining three domains (e.g. dvd, electronics, kitchen), and test on labeled data from the target domain. Furthermore, we follow Ziser & Reichart (2018) to evaluate adaptation into more distant domains; we train on all product reviews data and the unlabeled version of service reviews (i.e. the same documents without sentiment labels), then test on the labeled version of service reviews. Of all the training data used, less than $10\%$ are annotated with sentiment labels.

## 4.1 CROSS-DOMAIN SENTIMENT ANALYSIS

We compare our model with the following approaches: **1)** PBLM+CNN and PBLM+LSTM, which automatically construct a sentiment lexicon (i.e. pivots) from training data and use it to learn a Pivot-Based Language Model (PBLM); then, embeddings from the language model are fed to a CNN or LSTM for sentiment classification (Ziser & Reichart, 2018). We used the implementation by the authors[5] and ran it in our setting. **2)** ELMo+CNN, in which the deep contextualized ELMo embeddings[6] (Peters et al., 2018) are fed to a simple CNN text classifier. **3)** MAN[7] (Chen & Cardie, 2018), which proposes adversarial training techniques that can learn general and domain-specific features across multiple domains. **4)** VFAE[8], which extends a Variational Auto-Encoder model to handle domain adaptation (Louizos et al., 2015). **5)** GloVe+DAN, the original DAN implementation[9]

---

[4]Most previous works do not reduce duplicated documents, so the statistics for Multi-Domain Sentiment Dataset is 1000 documents, positive and negative each, per domain.

[5]https://github.com/yftah89/PBLM-Domain-Adaptation

[6]https://alpha.tfhub.dev/google/elmo/2

[7]https://github.com/ccsasuke/man

[8]https://github.com/NCTUMLlab/Huang-Ching-Wei

[9]https://github.com/miyyer/dan

| books | | | | trash | | error | |
|---|---|---|---|---|---|---|---|
| CBOW | jointCBOW | Ours $v_{\mathrm{senti}}$ | Ours $v_{\mathrm{domain}}$ | CBOW | Ours $v_{\mathrm{senti}}$ | CBOW | Ours $v_{\mathrm{senti}}$ |
| *novels* | *novels* | *novels* | *book* | *garbage* | *garbage* | *errors* | ***apology*** |
| *book* | *book* | ***movies*** | ***reading*** | *junk* | *junk* | *glitch* | *defect* |
| *articles* | *essays* | ***films*** | ***read*** | *crap* | *crap* | *defect* | *improper* |
| *essays* | *writings* | ***songs*** | *pages* | *rubbish* | *rubbish* | *email* | *agenda* |
| *writings* | *cookbooks* | *articles* | ***bookstore*** | *boston* | ***worthless*** | *h03* | *abandonment* |
| *poems* | *stories* | ***cds*** | ***authors*** | *vernacular* | ***horrible*** | *correction* | *oversight* |
| *novel* | *articles* | ***videos*** | ***author*** | *dreck* | ***useless*** | *inkling* | ***embarassment*** |
| *cookbooks* | *novel* | ***dvds*** | ***readers*** | *tripe* | *dreck* | *e-mail* | *abortion* |
| *magazines* | *prose* | ***cartoons*** | ***reader*** | ***dumpster*** | *sickness* | *anomaly* | *email* |
| *stories* | *texts* | *stories* | *novels* | *villages* | *drivel* | *ho3* | *assassination* |
| *textbooks* | ***movies*** | *magazines* | *lehane* | *o.c.* | *filth* | *headache* | *assembly* |
| *texts* | *poems* | *comics* | *mccullough* | *poop* | *landfill* | *stutters* | *activation* |
| *reviews* | ***animes*** | *programs* | *macomber* | *dung* | ***awful*** | *irq* | *abomination* |
| *manuals* | ***films*** | *book* | *calvino* | *lectroids* | ***nonsense*** | *notification* | ***unforgivable*** |
| *comics* | *magazines* | *cookbooks* | *robb* | ***excrement*** | *nov* | *abnormal* | *correction* |

Table 3: Top 15 similar words according to cosine similarity.

using the GloVe embedding[10]. **6)** DAN:DANN and CNN:DANN, in which we convert a DAN or CNN classifier into a Domain-Adversarial Neural Network (Ganin et al., 2016). **7)** Random+CNN and Random+DAN, supervised baselines trained from labeled data only; the input word vectors are randomly initialized.

The results are shown in Table 2. In each experiment, we ran our model 5 times with different random initialization and report the mean accuracy. The standard deviation is around $0.2 \sim 0.4\%$. Our sentiment classifier achieves high accuracy; it either outperforms previous state-of-the-art or ranks a close second[11], except for the `airport` domain where the sentiment labels are very unbalanced and the majority baseline can achieve an accuracy of $78\%$. In fact, by terms of F-score, our method outperforms PBLM+CNN and PBLM+LSTM in `airport` domain. Also, we significantly improve upon Random+CNN and Random+DAN, thus demonstrate the effect of semi-supervised learning. Further, we note that DAN is a strong baseline, as Random+DAN is competitive against several domain adaptation methods. We have also tried our implementation of Yoshida et al. (2011) and Wu & Huang (2016) in preliminary experiments, but they are not as good as GloVe+DAN.

## 4.2 SENTIMENT-AWARE WORD EMBEDDINGS

**How do our jointly trained, sentiment- and domain-aware embeddings differ from the CBOW model?** Qualitatively, we compare these vectors by assessing the 15 most similar words according to cosine similarity. In Table 3, we compare our model trained on all product reviews except the labeled data from `books` domain, and the CBOW embeddings trained on the same data. We first take the word "*books*" and see its $v_{\mathrm{senti}}$ more similar to other types of products such as "*films*" and "*songs*", compared to CBOW. Partially the reason is joint training with a CNN, as suggested by the jointCBOW column where CBOW is used as input to a CNN classifier and jointly trained, but without any sentiment enhancement or domain-focused part. We see jointCBOW slightly promotes "*films*" and "*animes*" but not as much as $v_{\mathrm{senti}}$. It might be because $v_{\mathrm{domain}}$ absorbs the domain specialty and enables $v_{\mathrm{senti}}$ to capture similarity across domains. Next, we take the words "*trash*" and "*error*" to confirm that $v_{\mathrm{senti}}$ emphasizes the sentiment or emotional aspect of meaning. For example, "*trash*" is similar to "*horrible*" in terms of $v_{\mathrm{senti}}$, but it is not the case in terms of CBOW. Similarly, "*error*" is similar to "*apology*" in terms of $v_{\mathrm{senti}}$.

**Can our embeddings distinguish sentiment polarity?** In Table 4, we take different context and polarity, and assess the 15 most likely cooccurring words (i.e., words whose $v_{\mathrm{senti}}$ have the largest dot products with the vector $M(y)\left(\sum_j u(w_j)\right) + b(y)$, where $w_j$'s are the context words and $y$ is the sentiment polarity). For the context "*is __ and*", the target words tha most likely to fill in the blank are

---

[10] https://nlp.stanford.edu/projects/glove/
[11] Bold values are significant ($p < .1$) assuming the test results follow Gaussian distribution.

| *is __ and* | | *is __ but* | | *__ the story* | |
|---|---|---|---|---|---|
| `positive` | `negative` | `positive` | `negative` | `positive` | `negative` |
| *fantastic* | *monotonous* | *fantastic* | *horrible* | *love* | *love* |
| *terrific* | *pointless* | *terrific* | *pointless* | *tells* | *tells* |
| *awesome* | *horrible* | *fine* | ***nothing*** | *tell* | *tell* |
| *amazing* | *absent* | *awesome* | ***ok*** | *telling* | *telling* |
| *superb* | *uneven* | *good* | *misleading* | *appreciate* | ***ruining*** |
| *fun* | *boring* | ***pricey*** | *outdated* | *true* | *into* |
| *breathtaking* | *unacceptable* | *amazing* | *monotonous* | *into* | *short* |
| *excellent* | *outdated* | *excellent* | *not* | *narrates* | *follows* |
| *cute* | *unbelievable* | *great* | *unacceptable* | *follows* | *behind* |
| *enthralling* | *weak* | *n't* | ***alright*** | *touching* | *true* |
| *wonderful* | *ineffective* | *superb* | *fine* | *throughout* | *narrates* |
| *beautiful* | *dumb* | *perfect* | *good* | *gripping* | *throughout* |
| *fabulous* | *terrible* | *nice* | *ridiculous* | *loved* | *about* |
| *perfect* | *inconsistent* | *not* | *uneven* | *short* | *thru* |
| *flawless* | *hysterical* | *cute* | *n't* | *behind* | *told* |

Table 4: Top 15 target words cooccurring with different context.

Figure 2: Heatmap of log-likelihood ratios indicating Bayesian inference of sentiment polarity. Blue denotes positive and red negative.

sentiment words that align to the `positive` or `negative` polarity. When context becomes more nuanced, e.g. "*is __ but*", some positive words appear under `negative` polarity, e.g. *ok* and *alright*; and vice versa, e.g. *pricey*. This suggests that our embeddings can model sentiment in different context. As for "*__ the story*", the target word *ruining* only appears under negative polarity. It is noteworthy that the embeddings presented here are trained without any labeled data from the `books` domain; still, they seem capture sentiment of phrases in book reviews.

**To further demonstrate that our embedding model scoops up sentiment from context,** in Figure 2 we show a heatmap of the Bayesian inference of sentiment polarity for each target word according to its surrounding context words (i.e., the color denotes $\log p(w \,|\, y = \texttt{positive}) - \log p(w \,|\, y = \texttt{negative})$ for each target word $w$, and the word generation probability $p$ depends on surrounding context, as given by Equation (5)). The documents are taken from unlabeled data in `books` domain. Note that the word "*good*" can be either positive (as in "*good for anyone*") or negative (as in "*expecting a good murder*"), according to context.

## 4.3 ABLATIVE TESTS

**Does better classifier lead to better Bayesian inference?** Since our model is formalized as a variational Bayes approximation, it is not obvious whether an approximation is necessary when precise Bayesian inference is possible, and whether combining a classifier with a label-enhanced embedding model is actually beneficial. To investigate, we evaluate the quality of our embedding model by the accuracy of its Bayesian inference of the sentiment polarity (i.e., using embeddings solely to classify sentiment, by calculating $\sum_{w \in D} \log p(w \,|\, y = \texttt{positive}) - \log p(w \,|\, y = \texttt{negative})$). The

| | books | dvd | electr. | kitchen | airline | airport | lounge | seat |
|---|---|---|---|---|---|---|---|---|
| Ours | 82.57 | 82.72 | **84.51** | 86.86 | **83.65** | 66.17 | 73.87 | **84.70** |
| Ours→Bayes | **83.76** | **83.53** | 82.59 | **87.31** | 75.54 | 72.20 | **74.93** | 83.01 |
| DAN | 77.11 | 77.22 | 77.67 | 78.08 | 73.12 | 68.99 | 72.85 | 76.95 |
| DAN→Bayes | 81.16 | 77.19 | 74.95 | 77.96 | 65.51 | **74.25** | 74.37 | 76.69 |
| Labeled→Bayes | 76.48 | 77.33 | 80.15 | 81.18 | 76.56 | 62.32 | 64.66 | 69.26 |

Table 5: Accuracy of sentiment classifiers and Bayesian inference.

| | books | dvd | electr. | kitchen | airline | airport | lounge | seat |
|---|---|---|---|---|---|---|---|---|
| Ours | **82.57** | 82.72 | **84.51** | 86.86 | **83.65** | 66.17 | 73.87 | 84.70 |
|   no domain | 81.41 | 83.16 | 81.77 | 85.33 | 79.18 | 70.12 | **74.93** | 83.93 |
|   joint CBOW | 79.78 | 82.55 | 81.95 | 85.20 | 78.19 | **70.53** | 74.68 | 82.78 |
|   no signal | **82.52** | **84.23** | **84.49** | **87.06** | 82.30 | 68.44 | 74.30 | **85.57** |
|   no DAN | 81.97 | 83.24 | 84.23 | 86.04 | 82.00 | 67.92 | 73.90 | 84.85 |

Table 6: Ablation of model components.

results are shown in Table 5. We see that the Bayesian inference of our embeddings (Ours→Bayes) generally achieves high accuracy, sometimes even outperforms our classifier. Next, we modify our classifier by replacing the CNN with a DAN. This leads to a weaker classifier (DAN), and we see that the accuracy of Bayesian inference (Ours→Bayes and DAN→Bayes) correlates perfectly with the jointly trained classifier (Ours and DAN). Further, we compare with the Bayesian inference using embeddings trained from labeled data only (Labeled→Bayes). It is worse than Ours→Bayes, which suggests that involving a classifier can actually improve upon pure Bayes. We also tried precise Bayes inference using both labeled and unlabeled data, but it did not work because the resulting embeddings tend to infer sentiment as either all `positive` or all `negative`.

**Do domain information and sentiment-aware modeling help?** In Table 6, we modify our model by removing the domain-focused embedding $v_{\text{domain}}$ and the domain-specific part of our classifier (no domain), or we further remove the sentiment-aware part of our embedding model (jointCBOW), and see the numbers decrease in most cases. It suggests that domain information and sentiment-aware modeling can indeed help embeddings improve sentiment classification.

**Can embeddings provide training signals to the classifier?** In our model, embeddings may help classifier as suitable input, or they may provide training signals through Bayesian inference on unlabeled data. In Table 5, Bayesian inference demonstrates its potential as training signal, as the accuracy sometimes surpasses the classifier. In Table 6, we changed the training of our model so that no update is back-propagated to the classifier $q_\phi(z \mid D, c)$ through the variational Bayes objective (no signal). To our surprise, the accuracy increases in several cases, suggesting that training signal from embeddings may not always help. Nevertheless, the training signal improves the classifier in one domain, `airline`, which is quite significant considering the large data size of `airline` domain and its distance from the labeled training domains (product reviews). Interestingly, the improvement will disappear if we remove the domain-specific part of our classifier (no DAN); it suggests that our embeddings help the classifier learn domain-specific tendency in `airline`.

## 5 CONCLUSION

We have shown that sentiment-aware embeddings can be trained from an unlabeled corpus and only a few labeled data, with the help of a sentiment classifier and improving that classifier in return. Moreover, by integrating domain information, the embeddings exhibit favorable generalization ability across multiple domains, and help adapt the sentiment classifier into completely new ones. Besides improving sentiment classification at the document-level, Figure 2 suggests that our trained embeddings might even help fine-grained aspect-level sentiment classification, a research direction that has come to interest recently (He et al., 2018).

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

## APPENDIX

Here, we provide details of our classifier model and the joint training strategy.

### CONVOLUTIONAL NEURAL NETWORK

Our CNN classifier takes $\boldsymbol{v}_{\text{senti}}$ as input; it scans every $l$ consecutive words $(w_i, \ldots, w_{i+l-1})$ in a document and concatenates their embeddings:

$$\boldsymbol{s}_i = [\boldsymbol{v}_{\text{senti}}(w_i) : \cdots : \boldsymbol{v}_{\text{senti}}(w_{i+l-1})]. \tag{11}$$

Then, a vector $\boldsymbol{x}_{\text{CNN}}$ is extracted by multiplying a filter matrix $\boldsymbol{R}$ and max-pooling:

$$(\boldsymbol{x}_{\text{CNN}})_j = \max_i (\boldsymbol{R}\boldsymbol{s}_i)_j. \tag{12}$$

Here, $(\cdot)_j$ denotes the $j$-th entry of a vector. Thus, every row of $\boldsymbol{R}$ sees all consecutive $l$ words and learns to select a distinctive phrase. Once $\boldsymbol{x}_{\text{CNN}}$ is obtained, we further apply a feed-forward layer and get the feature vector:

$$\boldsymbol{f}_{\text{CNN}}(D) = \text{ReLU}(\boldsymbol{W}_{\text{CNN}}\,\boldsymbol{x}_{\text{CNN}}). \tag{13}$$

In which, $\text{ReLU}(\cdot)$ denotes the Rectified Linear Unit. In this work, we fix the length of phrases to $l = 5$, and the dimensions of all feed-forward layers to 256.

### DEEP AVERAGING NETWORK

The DAN takes $\boldsymbol{v}_{\text{domain}}$ as input and extracts a vector $\boldsymbol{x}_{\text{DAN}}$ from document by averaging embeddings of all words:

$$\boldsymbol{x}_{\text{DAN}} = \frac{1}{n} \sum_{i=1}^{n} \boldsymbol{v}_{\text{domain}}(w_i). \tag{14}$$

Then, it simply applies multiple feed-forward layers:

$$\boldsymbol{f}_{\text{DAN}}(D; c) = \text{ReLU}\left(\boldsymbol{W}_{\text{DAN},m}(c) \cdots \text{ReLU}\left(\boldsymbol{W}_{\text{DAN},1}(c)\,\boldsymbol{x}_{\text{DAN}}\right)\right). \tag{15}$$

As a domain-specific part of our sentiment classifier, the parameters $\boldsymbol{W}_{\text{DAN},1}, \ldots, \boldsymbol{W}_{\text{DAN},m}$ here are domain-dependent. We set the number of feed-forward layers to $m = 3$, and their dimensions to 256.

Thus, our classifier has parameters $\phi = \{\boldsymbol{v}_{\text{senti}}, \boldsymbol{R}, \boldsymbol{W}_{\text{CNN}}, \boldsymbol{q}_{\text{gen}}, \boldsymbol{v}_{\text{domain}}, \boldsymbol{W}_{\text{DAN},1}, \ldots, \boldsymbol{W}_{\text{DAN},m}, \boldsymbol{q}_{\text{spec}}\}$. The embeddings $\boldsymbol{v}_{\text{senti}}$ and $\boldsymbol{v}_{\text{domain}}$ are shared between $\phi$ (the classifier) and $\theta$ (our embedding model).

### JOINT TRAINING TECHNIQUES

Due to the nature of our CBOW-like embedding model, the norms of embeddings tend to correlate with word frequencies. Our preliminary experiments suggest that large variation of embedding norms may harm the jointly trained classifier. Therefore, we always normalize the embeddings for training our classifier: Instead of directly using $\boldsymbol{v}_{\text{senti}}$ and $\boldsymbol{v}_{\text{domain}}$ in Equation (11) and Equation (14), we use the scaled $\tilde{\boldsymbol{v}}_{\text{senti}}$ and $\tilde{\boldsymbol{v}}_{\text{domain}}$ such that

$$\|\tilde{\boldsymbol{v}}_{\text{senti}}\|^2 + \|\tilde{\boldsymbol{v}}_{\text{domain}}\|^2 = 2. \tag{16}$$

Another issue with joint training is that, $\boldsymbol{v}_{\text{senti}}$ and $\boldsymbol{v}_{\text{domain}}$ receive updates from both the embedding model and the classifier. Our preliminary experiments suggest that, the norm ratio of these two types of updates may drastically affect the performance of the finally trained classifier. Therefore, we set different learning rates for different types of updates, such that "the norm of updates coming from the embedding model"/"the norm of updates coming from the classifier" is about $1/256$.

Furthermore, we set a smaller learning rate for updates coming to the classifier through the variational Bayes objective; thus for classifiers, "the norm of updates coming from unlabeled data"/"the norm of updates coming from labeled data" is adjusted to about $1/1024$.

