# OpenReview forum: "Semi-supervised Learning with Multi-Domain Sentiment Word Embeddings"
_ICLR.cc/2019/Conference_

### Official Review · AnonReviewer3 · 2018-10-30
**Review #3**

**Rating:** 6
**Confidence:** 3

**Review:**

This paper proposes to jointly train a classifier with a domain and label-aware word embedding model and a variational Bayes model for sentiment domain adaptation. The model is evaluated on a standard multi-domain sentiment analysis dataset where it achieves convincing results against strong baselines. Extensive ablations are conducted.

Pros:
- The paper is clearly written. I particularly appreciated Figure 1 as it might otherwise be difficult to see the relation of the different components of the model.
- The model achieves convincing results and ablations and analyses are extensive.

Cons:
- None of the presented ideas are entirely novel. The model rather combines many existing ideas successfully.
- The framework consisting of many components (particularly the joint training with CBOW as indicated in the appendix) seems rather brittle and very task-specific. I am concerned if this framework will be able to work on other tasks. I would love to see an evaluation on another dataset.
- The joint variational Bayes approach seems to be the most interesting aspect of the paper. Despite the ablations, it's not entirely clear to me, though, how useful this component is and if it can be applied beyond this particular model. I would like to see one of the baselines models or another model augmented with this component.

Questions:
- Do you alternate updates between each of the components or use a more sophisticated multi-task learning strategy when training the word embedding model and the other components jointly? Did you try fine-tuning the trained word embeddings with the classifier?
- Why do you use this particular affine transformation for learning sentiment-specific word embeddings? Did you try, for instance, an MLP as used in [1]?
- You say that you use a classifier q_φ(z |D, c) in order to benefit from more freedom of design for Bayesian inference. Could you elaborate why you use this classifier in addition to the label classifier q_φ(y |D, c)? In Table 6, it seems that it doesn't help much. The use of the term "classifier" is confusing at times, as it seems to be both used to refer to the latent variable and the label classifier.

[1] Tang, D., Wei, F., Yang, N., Zhou, M., Liu, T., & Qin, B. (2014). Learning Sentiment-Specific Word Embedding. Proceedings of the 52nd Annual Meeting of the Association for Computational Linguistics, 1, 1555–1565.

---

> ### Author Response · Authors · 2018-11-20
> **Thanks! and clarifying some details**
>
> Thanks for the helpful comment. Our answers are as below:
>
> > framework seems rather brittle and very task-specific
>
> It is true that the hyper-parameters have to be set correctly in order to train the model appropriately. But this, we believe, is an issue for all neural networks. In our case, models trained with suboptimal hyper-parameters are qualitatively different and quite easy to spot, for example by assessing top 15 similar words. Once the correct set of hyper-parameters is found, it seems working for all datasets: Note that we fix the same set of hyper-parameters for all experiments in this work, and qualitative inspection suggests that all trainings went well. Furthermore, at least some of the hyper-parameters are likely portable to other models/tasks, because it should be much the same as in this model we have inherited the same hyper-parameters we previously found effective for CBOW.
>
> > if variational Bayes can be applied beyond this particular model
>
> Actually, before we switched to the current CNN classifier, we have applied variational Bayes to a simple DAN classifier and confirmed that it can improve performance (eg., it improves books domain from 74% to 78%). Afterwards, we found that CNN better than DAN in this task, so we did not present the DAN results in the paper. But thanks for the suggestion!
>
> > Do you alternate updates between each of the components
>
> No, all components are jointly trained, meaning that they are updated at each step. We use an enhanced SGD optimization, in which learning rates are adjusted so that the norms of updates for each component are controlled during training (similar idea is found in Takahashi et al. ACL2018, and also controlling the norms of updates is the main ingredient of Adam).
>
> > Did you try fine-tuning the trained word embeddings with the classifier?
>
> Yes, we do fine-tune the word embeddings with the classifier. It is conducted jointly with the training of word embeddings. Details of this joint training are found in Appendix. A qualitative comparison showing the effect of joint training is found in Table 3.
>
> > Why do you use this particular affine transformation
>
> We believe an affine transformation is just an MLP reduced to one layer and without non-linear activation. We chose it because it is simple yet expressive enough to model interactions between sentiment and surrounding context. We did not try Tang et al. (ACL2014)’s model specifically because it cannot be trained on unlabeled data. But we will include and discuss it in Related Works.
>
> > q_φ(z |D, c) and q_φ(y |D, c)
>
> This might be a misunderstanding. q_φ(z |D, c) and q_φ(y |D, c) denote the same classifier; only that the classifier is used differently. When the sentiment label is not observed (i.e. unlabeled data), the classifier is used to PREDICT a probabilistic distribution of the sentiment label (i.e. q_φ(z |D, c)), and this “pseudo-label” is used for training embeddings. In this way our sentiment-aware embeddings can be trained on unlabeled data. On the other hand, when the sentiment label is observed, the classifier is TRAINED by maximizing log-likelihood of q_φ(y |D, c). We’ll try to make this clearer.

---

### Official Review · AnonReviewer1 · 2018-11-01
**Interesting approach and well conducted evaluation; some details missing and limited applicability**

**Rating:** 6
**Confidence:** 3

**Review:**

The authors propose a semi-supervised learning techhnique involving jointly tuning word embeddings and a classifier. The idea is to rapidly adapt models to new domains with minimal (actually, zero) supervision by exploiting such embeddings.

In effect, this is an approach to "disentangling" sentiment from domain, via a variational objective and semi-supervision via the embedding parameters. This is a nice idea as disentanglement enables transfer (or should). The experiments are well-executed (albeit on a limited set of classification tasks). Still the comparisons are relatively exhaustive, and the authors have done a nice job of providing ablations. I found Table 4 and Figure 2 particularly nice.

- The authors should speak to the generalizability of this approach beyond sentiment tasks, as presently it seems largely constrained to this domain. This would of course hinder the potential utility/impact of the work.

- Due to the very concise description of baselines, it was not obvious to me if these were also taking a semi-supervised approach? I *think* the pivot-based model is. But then how was CNN+ELMo trained exactly? With zero target sentiment labels, I presume? If so, an obvious baseline would be to "pseudo-label" instances in the target domain first, and then use these predictions as a target to fine-tune the CNN (or whatever), back-propping through to the embeddings. Was this done? In general more details on the baselines and training setups should be included, even if only in the Appendix.

- The strategy, I think, is to learn a prediction model p_\phi(y|z, D) where \phi are model parameters. Then this is applied to instances in the target dataset to infer latent z, and then use these inferred labels to fine-tune word embeddings. Specifically, sentiment label information is pushed into embeddings via a modified CBOW objective that effectively shifts the meaning (as codified in the distributed representation) via task specific sentiment. One thing here that confused me is that this would seem to perform this affine transformation to all words, but only certain of these will exhibit domain specific variation with respect to sentiment. Can the authors speak to this?

- I found Eq 7 counterintuitive, since it treats sentiment and domain independently, but the authors explicitly noted above that this is not the case, i.e., words will depend jointly on sentiment + domain. I actually think p(w_i|y) also depends on the domain via M (implicitly), but perhaps this should be made explicit (or perhaps I am misunderstanding something!)

- In 2.1, the authors assume a prior p(z|c) and then say that `'naively' marginalizing over this performs poorly. But I not think the particular prior distribution used was not discussed here. Could they elaborate?

- I think i indices are needed on the w's in Eq 6?

---

> ### Author Response · Authors · 2018-11-20
> **We will add broader discussion; about some details and an additional experiment**
>
> Thanks for the insightful review and suggestion of the word “disentangling”. It really puts this work in a broader context of representation learning.
>
> > The authors should speak to the generalizability of this approach beyond sentiment tasks
>
> Indeed. Thanks for the suggestion. We have conducted this work partially because it is a neat yet important application of semi-supervised learning by combining generative and discriminative models. This approach to semi-supervised learning is general enough to be applied to various tasks, and we see plenty of its potential in Natural Language Processing to overcome data sparsity and increase accuracy. For example, combining parsing and sentence generation might well be a fruitful direction. The generative model in this work has its own characteristics in that it disentangles sentiment from a domain specific part in order to enhance domain adaptation. We believe similar techniques can be useful in other tasks, given the ubiquity of word embeddings and the constant need for domain adaptation. However, we believe specific model designs should be considered task-by-task; frankly we do not believe a general model can always perform superbly in every task (recall the No Free Lunch Theorem). We will add more discussion about generalizability of this approach in our Conclusion, as well as Related Works being discussed with AnonReviewer2.
>
> > very concise description of baselines; CNN+ELMo with pseudo-labels
>
> We will add more details about the baselines. And yes, CNN+ELMo is trained with zero target sentiment labels. Actually, in our pre-experiments we have tried self-training with the DAN classifier; namely, using labeled data to train DAN in the first round, predict latent distributions for unlabeled data, then sample pseudo-labels for unlabeled data and mix them with labeled data, in order to train DAN in the second round. This approach brought almost no change to the performance (and in hindsight, drastic changes seem unlikely to us); so we abandoned this self-training idea. However, since the reviewer brought it up, we tried CNN+ELMo with pseudo-labels as indicated (the experiments just completed very recently). This time we did not sample pseudo-labels using predicted distributions, but simply use the predicted labels to annotate unlabeled data and train CNN+ELMo again. The results are as follows:
>
> Books: 77.58    Dvd: 78.56    Electronics: 82.09    Kitchen: 83.06
> Airline: 80.16    Airport: 64.69    Lounge: 66.46    Seat: 79.52
>
> As it shows, the results are better than Random+CNN but worse than ELMo+CNN trained on labeled data. We consider ELMo+CNN as a semi-supervised approach in a broad sense: namely, the large unlabeled corpus used for training the ELMo model certainly has effects.
>
> > affine transformation to all words
>
> Yes that’s correct. As a result, it is no surprise that the learned affine transformations for positive and negative sentiments are similar, because only a limited set of words exhibit variation w.r.t. sentiment. Take the model tested on books domain as an example (models for other domains are mostly the same): for each sentiment, if we decompose the learned matrix M into A+B, where A is a scalar multiplication of the identity matrix and B is orthogonal to A; then the Frobenius norms of A and B are about the same, and the cosine similarity between B_positive and B_negative is about 0.95. As for the learned vector b, the cosine similarity between b_positive and b_negative is about 0.93.
>
> > Eq 7 counterintuitive
>
> This might be a misunderstanding. Modeling the sentiment and domain parts as independent factors of word generation helps learn disentangled representations (and in this way word generation still depends on both sentiment and domain). On the other hand, it is indeed conventional wisdom that sentiment and domain interact with each other in word generation; but we believe this is more about the surrounding context than the domain per se. For example, it is said that “easy” is usually used as positive in electronics but negative in books domain, but it should be more suited to say that “easy” is used positively in the context “easy to use”, but negatively in “easy to guess ending”. Thus, we model sentiment interacting with surrounding context words instead of the domain vector. In this sense, yes, p(w_i|y) also depends on the surrounding context words, which implicitly carry domain information to some extent, but surrounding context itself is domain independent and we believe it is better to use than the domain vector. We will try to make this point clearer in the paper.
>
> > prior p(z|c)
>
> For the purely Bayesian approach, we have tried both setting prior p(z|c) to the uniform distribution, or learning it as specified in Section 2.4. In both cases the model learns to infer all-positive or all-negative labels. We will improve the narration and make this clearer.
>
> > i indices are needed on the w's in Eq 6
>
> That’s correct. We’ll fix it. Thanks!

---

### Official Review · AnonReviewer2 · 2018-11-01
**Nice Application of Variational Bayes semi-supervised learning**

**Rating:** 6
**Confidence:** 3

**Review:**

This paper jointly trains a sentiment classifier with a sentiment and domain-aware embedding
model, using both labeled and unlabeled data. When sentiment label is observed, this model is trained
with the usual cross entropy and maximum likelihood objectives; for unlabeled data, it uses pseudo
labels produced by the sentiment classifier with variational Bayes objective. This idea is not novel but the authors report that there is no previous work that jointly trains sentiment aware embeddings with a sentiment classifier specifically, and makes use of an unlabeled corpus to improve both. However, there are general and broader methods such as 'Toward Controlled Generation of Text' by Hu et al that apply semi-supervised techniques for generation (not classification) with specific constraints (sentiment, domain, etc). There are other recent methods such as 'Improving Language Understanding by Generative Pre-Training' by Redford et al that use the idea of generative pre-training with discriminative fine-tuning that are task-agnostic and achieve very good performance - how does the paper compare to this approach?
The experiments and analysis is very well written in the paper. Table 4 also shows very interesting, somewhat surprising results in the paper. The authors say that they will release the code and data for this technique which will be useful for the sentiment analysis research community.

---

> ### Author Response · Authors · 2018-11-20
> **We see plenty of potential in combining generative and discriminative models in NLP**
>
> Thanks for the informative comment. Hu et al.’s ICML2017 paper is especially a very interesting read. It is not directly comparable to this paper because, as the reviewer already pointed out, Hu et al.’s is about (sentence) generation but this paper is about (document) classification. Nevertheless, let’s have a deep discussion about these two papers.
>
> First, Hu et al.’s is insightful in that it models different semantic attributes as independent factors which control generation; our paper shares a very similar spirit by learning a disentangled sentiment part and a domain specific part for word embeddings. When the data is unlabeled, both papers use discriminators to produce pseudo-labels for training the generative model.
>
> But then, Hu et al.’s model propagates gradients of the discriminator directly back to the generator, using softmax annealing to circumvent the discreteness of text data; this technique is unnecessary in our setting, because the classifier in our model uses word embeddings (rather than words themselves) as input, and the embeddings are shared between classifier and the generative model. In other words, Hu et al.’s approach uses the discriminator to train the generator, whereas in our model the two are jointly trained.
>
> On the other hand, Hu et al. achieve semi-supervised learning of the discriminator by synthesizing data samples from the generator, whereas we use a variational Bayes objective on unlabeled data. Synthesized data only work if one has a good generator; we know that LSTM is a good generator for short sentences, but things are less clear for other types of objects, such as lengthy documents. Therefore, I would not say which one is more general and broader method; both Hu et al.’s and our work are meaningful extensions/applications of Kingma’s variational Bayes framework.
>
> From a broader perspective, there is no doubt that combining generative and discriminative models is a promising methodology toward semi-supervised learning, and we are interested in this direction because there are plenty of potential applications in Natural Language Processing. For example, one could combine parser and sentence generator, by parsing the generated sentences back to semantic structures to check if intended meanings are conveyed. This idea dates back to old days such as in Duan and White (ACL2014), and is still very active such as in Konstas et al. (ACL2017). Domain adaptation of sentiment classifiers is just another, simple but important application task, and there is novelty in applying variational Bayes methods here. We believe the knowledge obtained here will be helpful to other tasks as well. We will modify Related Works to discuss this broader point, including Hu et al. and Redford et al., as is also suggested by AnonReviewer1.
>
> As for Redford et al.’s in-progress work, we believe it belongs to the realm of pretraining unsupervised language models in order to enhance a wide range of classification tasks. We have already compared with one of its state-of-the-art, ELMo, and shown that we can outperform ELMo in this task, with much smaller unlabeled data. The big difference here, is that the representation learned by our generative model is aware of task-specific latent labels (i.e. sentiment), thus it is more task-oriented; in contrast, representations learned by pretraining unsupervised language models are task-agnostic. Our ablation tests clearly show that more task-oriented representations can improve performance on that task.

---

### Meta-Review · Area_Chair1 · 2018-12-14
**A reasonable but not very novel approach to semi-supervised sentiment classification**

**Confidence:** 4
**Recommendation:** Reject

**Metareview:**

Pros:

- The paper is well written

Cons:

- Not very novel

- Evaluation only on sentiment classification, whereas approaches applicable in broader context exists


- There are question re baselines (R3)

Neither reviewer was particularly enthusiastic about the paper, I believe, mostly because of the limited score and novelty.